# A high resolution extra-terrestrial solar spectrum determined from ground-based solar irradiance measurements

Julian Gröbner[1], Ingo Kröger[2], Luca Egli[1], Gregor Hülsen[1], Stefan Riechelmann[2], and Peter Sperfeld[2]

[1]Physikalisch-Meteorologisches Observatorium Davos, World Radiation Center, Dorfstrasse 33, 7260 Davos Dorf, Switzerland

[2]Physikalisch-Technische Bundesanstalt, Bundesallee 100, Braunschweig, Germany

*Correspondence to:* Julian Gröbner (julian.groebner@pmodwrc.ch)

**Abstract.** A high resolution extraterrestrial solar spectrum has been determined from ground-based measurements of direct solar spectral irradiance over the wavelength range from 300 nm to 500 nm using the Langley-plot technique. The measurements were obtained at the Izaña Atmospheric Research Center from the Agencia Estatal de Meteorologia (AEMET), Tenerife, Spain during the period 12 to 24 September 2016. This solar spectrum (QASUMEFTS) was combined from medium resolution (bandpass of 0.86 nm) measurements of the QASUME spectroradiometer in the wavelength range from 300 nm to 500 nm and high resolution measurements (0.025 nm) from a fourier transform spectroradiometer over the wavelength range from 305 nm to 380 nm. The Kitt Peak Solar Flux Atlas was used to extend this high resolution solar spectrum to 500 nm. The expanded uncertainties of this solar spectrum are 2% between 310 nm and 500 nm and 4% at 300 nm. The comparison of this solar spectrum with solar spectra measured in space (top of the atmosphere) gives very good agreements in some cases, while in some other cases discrepancies of up to 5% were observed. The QASUMEFTS solar spectrum represents a benchmark dataset with uncertainties lower than anything previously published. The metrological traceability of the measurements to the International System of Units (SI) is assured by an unbroken chain of calibrations leading to the primary spectral irradiance standard of the Physikalisch-Technische Bundesanstalt in Germany.

## 1 Introduction

Quantifying the spectral solar radiation penetrating the atmosphere is crucial to understand and quantify its interaction with the atmosphere, oceans and the surface. Especially the ultraviolet part of the solar spectrum drives the photochemistry of a number of important atmospheric trace gases such as ozone, nitrogen dioxide and hydroxyl and has significant effects on the terrestrial and aquatic ecosystems. While airborne or surface based measurements of solar radiation can be used to quantify these processes, often these measurements are not available and radiative transfer calculations are used instead. In the latter case, the solar extra-terrestrial spectrum is a necessary parameter which is needed to perform these calculations. Similarly, the determination of atmospheric constituents from remote sensing applications requires a very precise knowledge of the solar spectrum penetrating the atmosphere. In all cases, the uncertainty of the solar extra-terrestrial spectrum is one of the main components in the corresponding uncertainty budgets. Therefore reducing the uncertainty of the solar extra-terrestrial spectrum has direct and significant implications on a large number of atmospheric and environmental activities.

In the past 25 years, a number of satellite experiments have measured the solar extraterrestrial spectrum from space to avoid atmospheric absorption and scattering effects, especially at wavelengths shorter than 300 nm where ozone and oxygen in the atmosphere absorb all incident radiation (e.g. Cebula et al. (1996); Thuillier et al. (1997); Harder et al., (2009)). While pre-launch calibration and characterisation procedures reach very low uncertainties, once in space the possibilities of verifying or recalibrating such instruments become very challenging. As recent studies have demonstrated (e.g. Schöll et al. (2016)), the solar spectra measured from satellite platforms can differ significantly between each other, due in part to instrument degradation issues arising from the harsh space environment and the difficulties in accounting for possible instrument changes between the pre-flight calibration and their operation in space. In a strict metrological sense, such measurements cannot be considered traceable to SI since metrological traceability inherently requires the repeated demonstration of the uninterrupted traceability to primary standards which currently is not available to instruments located in space (https://www.nist.gov/traceability/supplementary-materials-nist-policy-review#internal_map).

Prior to the space age, solar irradiance measurements were performed from the surface, applying zero-airmass extrapolation techniques to derive the solar irradiance at the top of the atmosphere, e.g. Ångström (1969); Shaw (1983). In more recent years, spectral solar irradiance measurements using spectroradiometers (e.g. Bais (1997); Gröbner and Kerr (2001); Bolsée et al. (2014) or sunphotometers (Schmid and Wehrli (1995)) were used to determine the spectral solar spectrum in wavelength regions unaffected by strong atmospheric absorption features. These measurements were either used for atmospheric research, or for the validation of existing solar extraterrestrial spectra obtained from space measurements or from models of the sun (Fontenla et al. (2006); Shapiro et al. (2010)). While ground-based measurements of the solar irradiance have the disadvantage of needing to account for changing atmospheric conditions, the considerable advantage over space-based instruments is the possibility of recalibrating ground-based instruments and thereby validating and confirming their traceability to SI.

In this study we present ground-based direct spectral solar irradiance measurements obtained with the transportable reference double monochromator spectroradiometer QASUME and a high resolution Fourier Transform Spectroradiometer (FTS) over the wavelength range from 300 nm to 500 nm and from 300 nm to 390 nm respectively. A high resolution absolute extraterrestrial solar spectrum is then derived by applying the Langley-plot technique to the measurements of each instrument before combining them to a single high resolution solar extraterrestrial spectrum.

## 2 Instruments and methods

The measurements were performed at the Izaña Atmospheric Observatory (IZO) located on the island of Tenerife (Canary Island, Spain, 28.309 N, 16.499 W) from 12 to 24 September 2016. IZO is a high mountain station at an elevation of 2373 m above sea level (a.s.l) above a strong subtropical temperature inversion layer, which acts as a natural barrier for local pollution and low-level clouds. QASUME was installed on the roof of the measurement building at about 20 m above ground, while the FTS was operated on the ground.

## 2.1 QASUME

The transportable reference spectroradiometer QASUME consists of a double monochromator with a focal length of 150 mm and two 2400 lines/mm gratings resulting in a full width at half maximum (FWHM) of 0.86 nm. The whole system resides in a temperature controlled enclosure to allow outdoor operation under constant ambient conditions. The solar radiation is collected with a temperature stabilised diffuser connected via an optical fiber to the entrance slit of the monochromator. A portable lamp monitoring system allows the calibration of the whole system while being deployed in the field. A detailed description of the system can be found in Gröbner et al. (2005); Hülsen et al. (2016). A collimator tube with a full opening angle of 2.5 ° is mounted on an optical tracker to which the diffuser head can be fitted, allowing the measurement of direct solar spectral irradiance. A comprehensive uncertainty budget for global solar spectral irradiance measurements was discussed in Hülsen et al. (2016). Due to the fact that direct solar irradiance measurements are not affected by the directional response of the diffuser nor by the diffuse sky radiation distribution, the resulting expanded uncertainty is reduced from 3.1% for global solar irradiance measurements to 1.83% for direct solar spectral irradiance measurements in the spectral range 300 nm to 500 nm. The increase in measurement noise at short wavelengths below 305 nm becomes only relevant at large zenith angles above 75° which are not used in this analysis. Solar spectra were measured every 15 minutes from sunrise to sunset. QASUME was calibrated every day using a portable lamp monitoring system with a set of three 250 W tungsten-halogen lamps in order to verify its stability and demonstrate its traceability to SI. The calibrations were taken into account daily, and varied by less than $\pm 0.5\%$ over the course of the campaign, which has been taken into account in the uncertainty budget (see Hülsen et al. (2016)).

## 2.2 Fourier transform spectroradiometer

The transportable Fourier Transform Spectroradiometer (FTS) consists of a Bruker Vertex80 Fourier Transform Spectroradiometer with a customized fibre based entrance optics for direct spectral irradiance measurements, i.e. a collimator tube with a field of view of approximately $\pm 3.5°$. The instrument is installed in a temperature controlled transportable housing. The internal detector is a UG11-filtered GaP-photodiode covering the spectral range from 300 nm to 390 nm. The wavenumber resolution of the FTS was set to 2 cm$^{-1}$ resulting in a wavelength resolution of less than 0.025 nm for this wavelength range. Solar spectra were obtained approximately every 50 seconds. The wavelength scale of the FTS is inherently traceable to SI using a stabilised internal HeNe laser. The wavelength uncertainty is estimated to be equal or less than 0.01 nm. The radiometric calibration of the FTS was performed by a comparison with a calibrated spectroradiometer under natural sunlight condition. The entrance optic was mounted on an optical tracker together with a monitor-filterradiometer. The monitor filterradiometer is a temperature controlled UG11 filtered GaP-photodiode mounted within a collimator tube. The monitor correction turned out to be necessary to correct for the instability of the absolute scale of the FTS spectrum. The filter radiometer correction factor $f_{\text{FR}}$ is a scaling factor derived from the ratio of the measured filterradiometer current and the product of the measured radiometric corrected FTS spectrum and the spectral responsivity of the filterradiometer determined previously at PTB,

$$f_{\text{FR}} = \frac{U_{\text{FR}}/R_{\text{shunt}}}{\int s_{\text{FR}}(\lambda) \cdot E_{\text{rel,FTS}_{\text{corr}}}(\lambda) d\lambda}, \tag{1}$$

where $U_{\mathrm{FR}}$ represents the measured voltage corresponding to the photocurrent of the filter radiometer, $R_{\mathrm{shunt}}$ the shunt resistance, $s_{\mathrm{FR}}(\lambda)$ the spectral responsivity of the filter radiometer and $E_{\mathrm{rel,FTS_{corr}}}$ the radiometric corrected normalised relative spectral irradiance measured by the FTS. The instability of the FTS over time is then corrected by multiplying the raw FTS signal with this correction factor. The expanded uncertainty for measurements of the relative direct solar spectral irradiance was determined to be between 2% and 4% in the spectral range 310 nm to 380 nm. Below 310 nm the uncertainty of the FTS rises rapidly due to the low signal to noise ratio of the instrument.

## 2.3 Solar extra-terrestrial spectrum retrieval

The method used to retrieve the solar extra-terrestrial spectrum from ground-based measurements of direct solar irradiance uses the Beer-Lambert law,

$$I_\lambda = I_\lambda^0 R_{\mathrm{SE}} e^{-\tau_\lambda m}, \tag{2}$$

where $I_\lambda$ represents the solar irradiance measurement at wavelength $\lambda$, $I_\lambda^0$ the solar irradiance at the top of the atmosphere, $R_{\mathrm{SE}}$ the sun-earth distance normalised to 1 AU, $\tau_\lambda$ the total optical depth and $m$ the airmass. By taking the logarithm and expanding the optical depth $\tau$ and the airmass $m$ for the main atmospheric constituents gives,

$$\log I_\lambda = \log(I_\lambda^0 R_{\mathrm{SE}}) - \tau_\lambda^{\mathrm{O_3}} m_{\mathrm{O_3}} - \tau_\lambda^{\mathrm{R}} m_{\mathrm{R}} - \tau_\lambda^{\mathrm{aod}} m_{\mathrm{aod}}, \tag{3}$$

where the superscripts $\mathrm{O_3}$, R, and aod represent the atmospheric ozone, Rayleigh, and aerosol optical depth (AOD) respectively. The representative airmass $m_{\mathrm{O_3}}$ for the atmospheric ozone is calculated assuming that all the ozone is concentrated in a layer at 22 km height, while $m_{\mathrm{R}}$ and $m_{\mathrm{aod}}$ are calculated for a layer at 5 km.

The Langley-plot technique consists in linearly regressing Eq. 3 versus airmass (in this study the airmass range from 1.1 to 3.5 was used) to retrieve the intersect $I^0$ for every wavelength separately, assuming that $\tau$ remains constant during this period. For this assumption to be valid, measurements are usually performed at high altitude sites where the aerosol optical depth is very small and variations in AOD will have no significant effect on the regression.

Due to the strong absorption of ozone below 340 nm, the regression in that wavelength region is done against $m_{\mathrm{O_3}}$, while at longer wavelengths the regression is done against $m_{\mathrm{R}}$. Since the Rayleigh optical depth $\tau_{\mathrm{R}}$ has a similar magnitude as the ozone optical depth at wavelengths below 340 nm, Eq. 3 is slightly rearranged to take into account the Rayleigh optical depth which can be calculated easily from the known atmospheric pressure at the measurement site,

$$\log I_\lambda + \tau_\lambda^{\mathrm{R}} m_{\mathrm{R}} = \log(I_\lambda^0 R_{\mathrm{SE}}) - \tau_\lambda^{\mathrm{O_3}} m_{\mathrm{O_3}} - \tau_\lambda^{\mathrm{aod}} m_{\mathrm{aod}}. \tag{4}$$

Figure 1 shows the log of the QASUME direct irradiance measurements at 310 nm, 320 nm, and 350 nm for the morning of 14 September 2016 versus airmass, from which the zero-air mass solar irradiances are retrieved by a linear extrapolation. In this particular example 18 measurements are used for this Langley-plot, while for the FTS, up to 450 points are used due to the higher sampling rate. The slopes of the linear curves represent the optical depth $\tau_\lambda^{\mathrm{O_3}} + \tau_\lambda^{\mathrm{aod}}$, which is becoming larger at short wavelengths due to the increase in ozone absorption with decreasing wavelength.

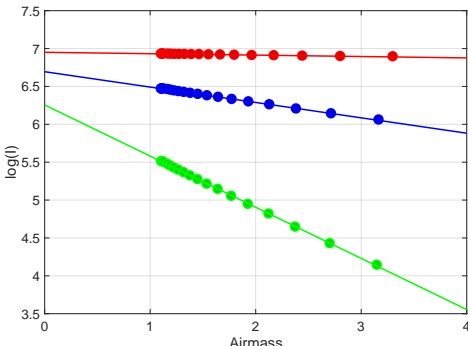

**Figure 1.** Langley plot using direct solar irradiance measurements of QASUME on the morning of 14 September 2016. The green, blue and red dots represent measurements at 310 nm, 320 nm, and 350 nm respectively. The lines represent a linear fit to the data points to retrieve the intercept at airmass 0.

At IZO, the aerosol optical depth at 500 nm is often smaller than 0.02; correspondingly the aerosols have a negligible contribution to the total optical depth in the wavelength range of strong ozone absorption, justifying the regression of Eq. 4 versus $m_{O_3}$ at wavelengths shorter than 340 nm. The Langley-plot procedure is quite robust for the atmospheric conditions found in September at IZO during the measurement campaign since most days have AOD variations of less than 0.005 at 500
nm and the total column ozone variations determined with co-located Brewer spectrophotometers are less than 2 DU during each Langley-plot period (half-day). Table 1 shows a summary of the atmospheric conditions encountered during the campaign which were used to select the most appropriate periods for the subsequent analysis.

The criteria used for an objective selection of measurements to include in the analysis were: total AOD less than 0.02, AOD variation equal or less than 0.005, and TCO variations less than 2 DU. For the measurement period, 7 out of a total
of 13 half-days satisfy these criteria, from which zero airmass solar spectra $I_0$ were retrieved. The ratio between these seven spectra to their average is shown in Figure 2 for QASUME. The variability observed at wavelengths shorter than 320 nm can be clearly correlated to the ozone variability during each Langley-plot period. Indeed, the regression of the residuals shown in Figure 2 at wavelengths shorter than 320 nm versus the observed ozone variation (not shown) gives a zero-crossing of the residuals for zero ozone variation, which coincides with the $I_0$ retrieved from the seven individual spectra, supporting the
assumption that the ozone variations do not introduce a significant systematic bias to the Langley-plot procedure. Furthermore, for wavelengths longer than 305 nm the solar extraterrestrial spectrum (ETS) retrieved from averaging the spectra from all 13 half-days as shown in Table 1 is only 0.2% higher than the solar ETS from the best seven half-days, demonstrating that the selection introduces no systematic bias to the retrieved $I_0$, but only decreases its standard deviation.

Due to the finite resolution of 0.86 nm of QASUME, a small spectral correction needs to be applied to the retrieved $I_0$ as
described in Gröbner and Kerr  (2001). This bandwidth correction is estimated by determining $I_0$ from synthetic (modelled)

**Table 1.** Summary of the atmospheric conditions during the measurement campaign. For each half-day (indicated by M (morning) and A (afternoon) the aerosol optical depth at 500 nm, the total column ozone (TCO) in DU and the change of AOD and TCO during the period is shown. The * indicates that this half day satisfied the criteria for inclusion in the average set.

| Day | AOD at 500 nm | dAOD | TCO | dTCO |
|-----|---------------|------|-----|------|
| *14M | 0.009 | -0.002 | 286.9 | 1.0 |
| *14A | 0.008 | 0.003 | 284.1 | 1.9 |
| 16M | 0.009 | -0.002 | 284.6 | -3.9 |
| *16A | 0.008 | 0.002 | 283.9 | -1.7 |
| *17M | 0.013 | 0.005 | 282.0 | -0.3 |
| 17A | 0.021 | 0.008 | 279.7 | -1.1 |
| 19M | 0.033 | 0.007 | 272.5 | -1.0 |
| 20M | 0.014 | 0.006 | 270.0 | -0.7 |
| 20A | 0.015 | 0.002 | 272.3 | 3.1 |
| 21M | 0.013 | -0.008 | 278.1 | 2.7 |
| *21A | 0.012 | 0.002 | 277.0 | -0.4 |
| *24M | 0.009 | 0.001 | 279.1 | -0.8 |
| *24A | 0.009 | 0.001 | 278.4 | 0.2 |

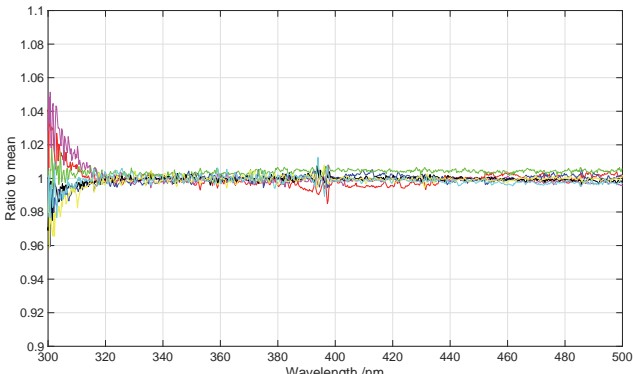

**Figure 2.** Ratio between solar spectra obtained from direct irradiance measurements with the QASUME spectroradiometer to their average during 7 half-days in the period 14 September to 24 September 2016.

direct solar irradiance spectra using a total column ozone value of 280 DU representative for the campaign and for an airmass range between 3.5 and 1. The high resolution solar ETS COKITHQA (Egli et al. (2012)), $I_0^{\text{ref}}$ based on the Kitt Peak Solar Flux Atlas (Kurucz et al. (1984)), is used as reference for these model calculations. The synthetic solar spectra are then convolved with the slit function of QASUME before determining $I_0^{\text{mod}}$ using Equation 4, thereby simulating the measurements
5   of QASUME during the campaign. The bandwidth correction which needs to be applied to the solar extraterrestrial spectra

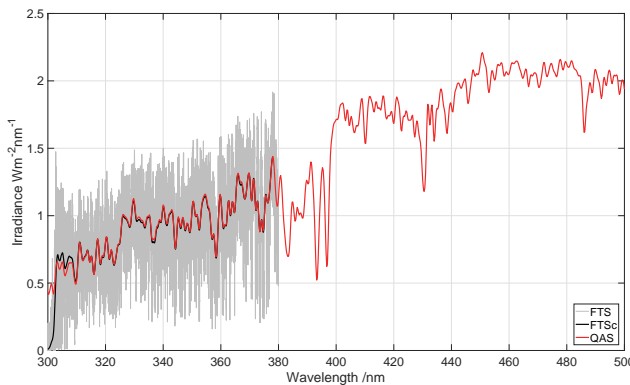

**Figure 3.** Solar spectra obtained from direct irradiance measurements with QASUME (red curve) and the FTS (grey curve). The black curve is calculated from convolving the high resolution spectrum of the FTS with the QASUME slit function.

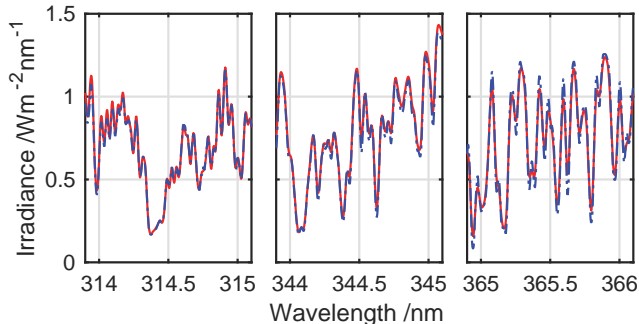

**Figure 4.** FTS (red curve) and COKITHQA (Kitt Peak) (dashed blue curve) solar spectra for three spectral intervals. The left figure shows the two solar spectra from 313.9 nm to 315.1 nm, the middle figure from 343.9 nm to 345.1 nm, and the right figure from 364.9 nm to 366.1 nm.

determined with QASUME is then simply the ratio between $I_0^{\mathrm{ref}}$ convolved with the QASUME slit function and $I_0^{\mathrm{mod}}$. The correction is negligible at wavelengths longer than 330 nm, and gradually increases to 4% at 300 nm, which is slightly larger than the correction shown in Gröbner and Kerr (2001) for a Brewer spectroradiometer with a resolution of 0.57 nm.

5     The same Langley-plot regression was applied to the direct irradiance measurements of the FTS. The measurements which were used for the analysis were obtained on 24 September, for a resolution setting of $2\,\mathrm{cm}^{-1}$ which is equivalent to a resolution of close to 0.025 nm in the measured wavelength range from 300 nm to 390 nm. Figure 3 shows the solar extraterrestrial spectra derived by QASUME and the FTS, as well as the solar spectrum of the FTS convolved with the QASUME slit function.

    The two solar spectra agree remarkably well, as can be seen in Figure 3. The largest differences arise at short wavelengths below 310 nm due to the low sensitivity of the FTS at larger airmasses which affect the retrieval of $I_0$ using the Langley-plot

10 procedure. Below 305 nm, the spectrum decreases significantly and does not give consistent results anymore.

We have compared the spectral features of the high resolution FTS spectrum shown in Figure 3 with the COKITHQA solar spectrum, whose high resolution component is based on the Kitt Peak Solar Flux Atlas. Due to the extremely high resolution of this solar spectrum, we have convolved it first with a 0.025 nm bandpass to bring it to the same resolution as the measurements of the FTS. We have furthermore selected three spectral regions of about 1 nm each to show the two solar spectra. Figure 4 shows the solar spectra of the FTS and Kitt Peak centered at 314.5 nm, 344.5 nm, and 365.5 nm. As can be seen in the figure, both spectra agree remarkably well with each other, both spectrally and in absolute levels. Specifically, we would like to emphasise the nearly perfect wavelength agreement between the two spectra, which was assessed by shifting one spectrum by a small wavelength increment and calculating the residuals of the spectral ratio between the two solar spectra. The lowest residuals were obtained for no shift, and already a relative wavelength shift of $\pm 1$ pm led to significantly larger residuals, confirming the perfect wavelength scale agreement between the FTS and Kitt Peak solar spectra.

## 2.4 Uncertainty estimation

The main contribution to the uncertainties of the solar extraterrestrial spectra shown in Figure 3 is the absolute calibration of the QASUME and FTS spectroradiometers. Additional uncertainty components are the variability of the Langley-plot retrievals (see Figure 2 for QASUME), the uncertainty in the finite bandwidth correction, as well as the uncertainty of the ozone and aerosol airmasses used in the Langley-plot retrievals. The uncertainty in the finite bandwidth correction is mainly a function of the total column ozone, which varied by $\pm 5$ DU during the campaign, and the effective height of the ozone layer above Izaña, $22 \text{ km} \pm 0.5 \text{ km}$ which was measured several times during the campaign by ozone sondes.

The scattered circumsolar radiation from Rayleigh scattering as well as from the forward scattering of aerosols can introduce a bias in the measurement of direct solar irradiance which is not accounted for by the simple Beer-Lambert law presented in Eq. 3. To estimate the magnitude of this effect for the conditions encountered in Izaña during the campaign, we estimated this contribution using the radiative transfer model libRadtran (Mayer and Kylling (2005)), using as basis for the radiative transfer calculations the atmospheric conditions found at Izaña, namely atmospheric pressure of 772 mbar and aerosol optical depth of 0.02 with an assumed aerosol asymmetry factor of 0.76. The model calculations were carried out for several airmasses between 1 and 3.5 and for the wavelength range between 290 nm and 500 nm, in order to simulate the effect of the scattered radiation on the Langley-plot retrieval. The contribution of scattered radiation in the field of view of QASUME with respect to the direct irradiance is largest at short wavelengths and large airmasses: at an airmass of 3.5 the relative contribution is 0.55% and 0.14% at 300 nm and 310 nm respectively. At smaller airmasses and longer wavelengths, this contribution decreases to less than 0.1%. The significant dependence on wavelength suggests that the dominant contribution comes from Rayleigh scattering, while the forward scattered radiation from the aerosols is negligible. This relative contribution is very small, and clearly within the measurement uncertainties. When the zero-airmass extrapolation is performed on the measurement data including the modeled relative contribution from circumsolar radiation, the resulting zero airmass spectrum differs by less than 0.1% from the original, demonstrating that scattered radiation in the field of view of QASUME has a negligible effect on the resulting extra-terrestrial solar spectrum and can therefore be neglected. While this effect is expected to be approximately a factor of 3 larger for the FTS due to its larger field of view with respect to QASUME, we have not taken it explicitly into account since any resulting

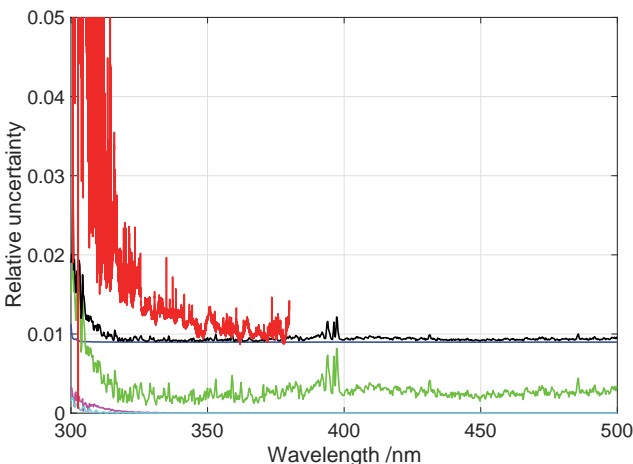

**Figure 5.** Relative uncertainty of the solar extraterrestrial spectra derived from the direct solar irradiance measurements using the Langley-plot method. The red and black curves represent the total relative uncertainties of the FTS and QASUME respectively. The remaining curves represent the uncertainty contributions calculated for QASUME for the direct solar irradiance measurements (blue), the Langley-plot variability (green), the effective ozone height (magenta) and the bandwidth correction (cyan).

bias in the retrieved solar spectrum of the FTS will be taken into account when the QASUME and FTS spectra are combined, as explained in the following section.

The individual uncertainty components are shown in Figure 5 for QASUME, as well as the combined uncertainties of QASUME and the FTS.

5  In order to produce a consistent high resolution solar extraterrestrial spectrum from QASUME and the FTS, and taking into account the smaller uncertainties of QASUME with respect to the FTS, the spectral ratio between QASUME and the convolved FTS spectrum in the wavelength range from 304.8 nm to 378 nm was applied to the high resolution FTS solar spectrum to normalise it to the QASUME absolute irradiance. This effectively transfers the absolute irradiance scale of QASUME to the high resolution solar spectrum measured with the FTS. Furthermore, the high resolution Kitt Peak Solar Flux Atlas was used

10  in conjunction with the QASUME solar spectrum for the wavelength range 300 nm to 304.8 nm and from 378 nm to 500 nm in order to extend the high resolution solar spectrum over the whole wavelength range of QASUME. The resulting high resolution solar spectrum QASUMEFTS combines the high resolution solar spectrum obtained with the FTS with the absolute irradiance scale of QASUME. The resulting expanded uncertainty (at 95% coverage probability) shown in Figure 5 for the combined QASUMEFTS solar spectrum is 2.0% between 310 nm and 500 nm, gradually increasing to 4% at 300 nm.

15  ## 3   Comparison to literature solar extraterrestrial spectra

This high resolution solar spectrum QASUMEFTS was compared to seven solar spectra determined either in space or through combination of the Kitt Peak Solar Flux Atlas with lower resolution space-based solar spectra.

From top to bottom, Figure 6 shows the comparison to the solar ET spectra measured with SOLSPEC onboard the ISS (the version SOLSPEC_A by Bolsée et al. (2017) and SOLSPEC_B by Meftah et al. (2016)), SIM version V17 onboard SORCE (Harder et al. (2010)), SCIAMACHY (Hilbig et al. (2017)), the ATLAS-3 solar spectrum (Thuillier et al., (2003)), the Chance-Kurucz solar spectrum (Chance and Kurucz, 2010) (SAO), a composite solar spectrum made up from a variety of spectra

(Gueymard (2003)), and finally another composite solar spectrum corrected by a comparison to ground-based measurements (Egli et al. (2012)). The wavelength scale of the five space-based solar spectra were adjusted to air wavelengths to provide a consistent wavelength scale for this comparison. The relative uncertainties of the space-based solar spectra are described in the respective references, while the two composite solar spectra only provide uncertainty estimates based on the spectra used to produce the composites. Unfortunately not every spectrum has a documented uncertainty budget analysis, and often

it is not clearly defined what coverage probability is used when quoting an uncertainty. Therefore only the expanded relative uncertainty of QASUME is shown in Figure 6.

With the exception of the SIM and SOLSPEC solar spectra where the slit function was available, the comparison was performed by convolving the spectra with a 1 nm full width at half maximum triangular slit function before calculating the ratio to QASUMEFTS, which was also convolved the same way. For SIM and SOLSPEC, the high resolution QASUMEFTS

spectrum was convolved with the spectrally varying instrument slit function before taking the ratio to the respective solar spectrum of the instrument.

The SORCE spectrum used here was measured on 27 November 2004 and the SCIAMACHY spectrum on 27 February 2003. SOLSPEC-A represents the average over the period 6 June 2008 to 29 April 2009 while SOLSPEC-B represents the average for the period April to June 2008. Finally the ATLAS-3 Spectrum was measured on several days in November 1994.

Thus all solar spectra shown here , including QASUME were either measured during solar minimum or in the declining phase of the respective solar cycle.

Current studies provide only an upper limit of the solar spectral irradiance (SSI) variability throughout a solar cycle of the order of 1% to 2%. Based on the results from SORCE SIM discussed in Harder et al., (2009), the variability of SSI between 300 nm to 500 nm is at most 0.5%. In that respect, the different measurement periods of the solar spectra discussed here should

not have a significant impact on the comparisons.

As can be seen in Figure 6, most of the solar spectra agree fairly well with QASUMEFTS. While SOLSPEC_A shows an overall agreement to QASUMEFTS within the uncertainties, SOLSPEC_B underestimates it by nearly 5%. A small discontinuity is observed at 320 nm with SOLSPEC_A, while a more smooth deviation is observed with SOLSPEC_B, which could be due to a filter change which was treated differently in the two analyses (personal communication, D. Bolsée).

The comparison to SORCE SIM shows good agreement to better than 2% at wavelengths longer than 340 nm, while at shorter wavelengths SORCE SIM gradually underestimates QASUMEFTS by up to about 5% at 310 nm.

The SCIAMACHY solar spectrum shows very good agreement over the whole wavelength range, with a small discontinuity at 395 nm, which comes from switching from one spectroradiometer channel to the next, which also includes a change of spectroradiometer resolution from about 0.2 nm to 0.4 nm (personal communication, M. Weber). The low spectral residuals

(black curve in Figure 6) show that the wavelength scale between SCIAMACHY and QASUMEFTS agrees very well, and that the resolution of SCIAMACHY is sufficient to allow the convolution with a 1 nm bandpass.

The ATLAS-3 Spectrum gives the overall best agreement with QASUMEFTS, being within the uncertainties of QA-SUMEFTS over the whole wavelength range for the 10 nm smoothed spectrum. The noisy residuals are an indication that the spectral resolution of the measurements is slightly too low for the 1 nm convolution, and that possibly some wavelength discrepancies between QASUMEFTS and ATLAS-3 might exist. The 10 nm running average smooths out the high frequency spectral noise and shows that on an absolute scale both spectra agree to better than 2% between 300 nm and 500 nm.

The comparisons to the three composite high resolution spectra (SAO, GUEYMARD and COKITHQA) show much less noise because these are also based on a high resolution spectrum and thus the convolution works very well. It also shows that in the region 305 nm to 380 nm the Kitt Peak Solar Flux Atlas and the high resolution spectrum obtained with the FTS agree very well. The larger spectral noise of COKITHQA than the one of GUEYMARD below 330 nm indicates that the procedure to create this composite needs to be improved in that wavelength region. While Gueymard (2003) uses ATLAS-3 for the wavelength region shorter than 400 nm, COKITHQA uses the ATLAS-3 solar spectrum as basis, together with minor modifications based on a comparison with ground-based measurements, as described in Egli et al. (2012).

Several spectral regions of the composite spectrum of GUEYMARD show deviations with respect to QASUMEFTS which are larger than the uncertainties. A general over-estimation of 1.1% between GUEYMARD and QASUMEFTS is observed, with differences of up to +3% in the region between 300 nm and 330 nm, and between 420 nm and 460 nm. The agreement of QASUMEFTS with COKITHQA is slightly better, with the largest deviations of +3% between 310 nm and 320 nm. For the remaining wavelength range, COKITHQA is within the uncertainties of QASUMEFTS, which shows the improvements obtained using ground-based measurements to correct the COKITHQA composite.

## 4   Conclusions

A high resolution extraterrestrial solar spectrum in the wavelength range from 300 nm to 500 nm has been determined using ground-based measurements of direct solar irradiance and applying the Langley-plot technique. This QASUMEFTS solar spectrum was constructed by combining the high resolution direct solar irradiance measurements of a fourier transform spectroradiometer with the moderate resolution of the QASUME spectroradiometer. The Kitt Peak Solar Flux Atlas was used for the wavelength range not covered by the FTS.

A comprehensive uncertainty budget evaluation for the derived solar spectrum has been determined, with expanded uncertainties ($k$=2, assuming a 95% probability coverage) of 2% between 310 nm and 500 nm. The uncertainties gradually increase to 4% at 300 nm, reflecting the increased uncertainties due to atmospheric ozone variations affecting the Langley-plot retrieval of the zero airmass solar spectrum at this short wavelengths, as well as the uncertainty of the finite bandwidth correction of the QASUME spectroradiometer.

The QASUMEFTS solar spectrum represents a benchmark dataset with uncertainties lower than anything previously published. The absolute irradiance scale of QASUME was monitored daily on-site using a portable field calibrator to monitor the

sensitivity of the instrument. The metrological traceability of the measurements to SI is therefore assured by an unbroken chain of calibrations leading to the primary spectral irradiance standard of the Physikalisch-Technische Bundesanstalt in Germany.

The comparison of the QASUMEFTS solar spectrum with space-based solar spectra has shown good agreement within the uncertainties in some cases, while in some other cases differences as large as 5% were observed.

This high resolution solar spectrum will also be useful to radiative transfer modeling of the atmosphere for calculating the solar radiation in the atmosphere and at the surface. Furthermore, direct solar irradiance measurements in conjunction with QA-SUMEFTS will eventually allow fully traceable atmospheric trace gas retrievals with a comprehensive uncertainty budget integrating not only measurement uncertainties but also model uncertainties from the retrieval algorithm, which is a major objective of the European Metrology Research Programme (EMRP) Joint Research Project ATMOZ (http://projects.pmodwrc.ch/atmoz/).

## 5  Data availability

The high resolution extraterrestrial solar spectrum QASUMEFTS can be obtained by anonymous ftp
at ftp://ftp.pmodwrc.ch/pub/people/julian.groebner/QASUMEFTS.dat

*Author contributions.*  JG, LE and GH operated the QASUME spectroradiometer, IK developed the FTS and SR and PSp operated the FTS during the campaign.

*Acknowledgements.*  NSO/Kitt Peak FTS data used here were produced by NSF/NOAO. The data is available from ftp://vso.nso.edu/pub/Kurucz_1984_atlas/. This work has been supported by the European Metrology Research Programme (EMRP) within the joint research project EMRP ENV59 ATMOZ "Traceability for atmospheric total column ozone." The EMRP is jointly funded by the EMRP participating countries within EURAMET and the European Union. J. Harder provided the SORCE SIM V17 data and helped in the data comparison of the SIM spectrum. The authors would also like to thank the staff of the Izaña atmospheric Observatory for their support during the campaign.

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

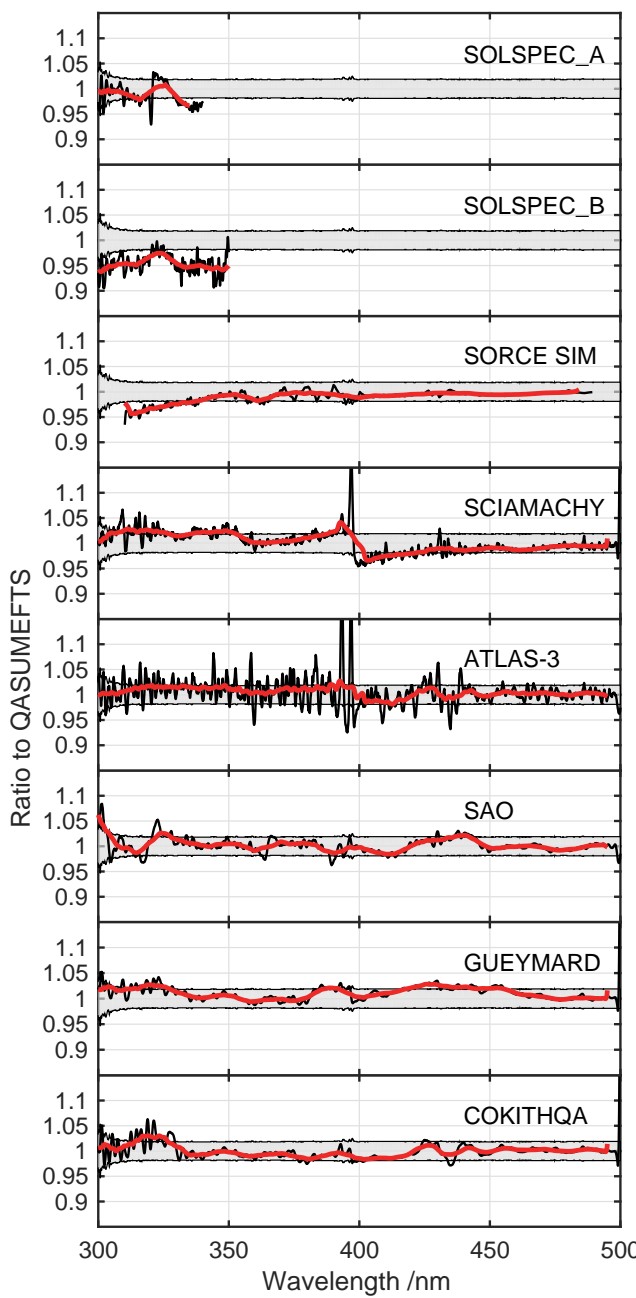

**Figure 6.** From top to bottom: Ratios between QASUMEFTS and the solar ET Spectrum from SOLSPEC (SOLSPEC_A by Bolsée et al. (2017) and SOLSPEC_B by Meftah et al. (2016)), SORCE SIM (Harder et al. (2010)), SCIAMACHY (Hilbig et al. (2017)), ATLAS-3 (Thuillier et al., (2003)), SAO (Chance and Kurucz, 2010), the composite from Gueymard (2003) and the composite from Egli et al. (2012). The black lines are the ratios convolved with a 1 nm wide triangular slit function, while the red line is a 10 nm running average. The gray shaded area represents the expanded uncertainty of the QASUMEFTS spectrum.