# Peer review of "The high resolution extra-terrestrial solar spectrum (QASUMEFTS) determined from ground-based solar irradiance measurements"

_Atmospheric Measurement Techniques, 2017_

## Referee Comment (RC1) · Anonymous Referee #1 · 29 May 2017

The manuscript presents a new high resolution (0.02 nm) extraterrestrial solar spectrum over the wavelength range 300 – 500 nm with an expanded uncertainty of 2% above 310 nm and 4% below. It is derived from ground based measurements of direct solar irradiance by 2 instruments, one with a very high absolute accuracy and the other with very high spectral resolution. To my knowledge this is the best available data at the moment, and therefore it is very useful for many applications in atmospheric sciences. I think this manuscript is a clear and significant step forward and therefore it is very worthwhile to be published in AMT.

The manuscript is very well written and clearly structured. Only a few specific points should be clarified by the authors prior to final publication:

p.3, ln. 3: it would be better to use instead of 'resolution' the more specific term 'full width at half maximum'. p.3, ln.7 and ln. 20: the field of view should be stated both times either with '+-' or with the full opening angle. p.3: for both instruments, what was the frequency of the measurements, how many data points are available for the fit on each half day? p.7, Fig. 3: the scales of the irradiance should be given for the 3 panels.

---

## Referee Comment (RC2) · Anonymous Referee #2 · 7 Jun 2017

**General comments**

This is a paper of high interest. Using only a well-proved technique (Langley) and high accurate, well calibrated instrumentation, the authors address a relevant scientific topic: an accurate estimation of extra-terrestrial Solar Spectral Irradiance (SSI), here presented between 300 and 500 nm. This is a major inputs for atmospheric science, obtained without any measurements from space, and also, an improvement of their own previous results (JGR, 2001). The spectral measurements at high resolution presented here meet the demands of the radiative transfer modelling because of the wavelength dependence of the photochemical reactions taking place in the planetary atmospheres. The problem of the limited spectral resolution of double monochromators is well addressed here by this combination of two instruments for the Langley plots. The quality of the results (instrumentation, scientific work, field measurements) is clearly sufficient to support the interpretations and conclusions, in spite of the cut-off at 300 nm (in comparison with space based measurements going down to the deep UV, but suffering in other hands, from many limitations to maintain the radiometric absolute scale). In the present paper, the results are clearly useful for atmospheric, ground-based and oceanic researches, and less for solar physics and SSI variability (because of the limitation to 300 nm). The paper present an original and new contribution, through fruitful collaborations between renowned institutes (PMOD, PTB, ISO), and the reference to previous an similar works is also well stated (text, number and quality of appropriate references). The traceability to the PTB absolute radiometric scale was well managed during all the duration of the field campaign.

The title is clear and reflect the content of the paper. The abstract present a concise summary. For the overall presentation, it is well structured but the balance between the text and the number of equations/plots could be a little bit revised, to avoid that mathematical formulae were sometime only described by sentences. For the reader, facing to the description to the instrumentation and data processing, it could be preferable to have more plots, schemes and equations. The general measurement equations (for both instruments) could be presented in the paper to provide a good general overview of the data processing. Some part of the paper could be clarified, as it will be explained here after.

**Specific comments**

- Page 3, line 8: description of the tube for direct solar measurements (FOV of 2.5°). Is there any estimation of the circumsolar contribution (2.5° minus the solar disk) that should be non-negligible in the UV during field measurements (in comparison to space measurements)? At the PTB, the straylight in this FOV should have been normally removed during the calibration. Maybe the stability and performances of the solar tracker allowed a possible reduction of the FOV to ~1°? It is an important topic for the extra-terrestrial SSI retrieval from Langley plots.

- Page 3, line 14: only one lamp was used to monitor a possible change of absolute responsivity due to transportation and aging effects. Why not more than one lamp? A triplet of lamps would have improved the uncertainty budget.

- Page 3, line 20: same remark for the FOV and circumsolar UV contribution, fully negligible?

- Page 3, lines 25-29: it could be better for the reader to present the equation of this instability correction, instead of sentences. We understand that a comparison between the filter radiometer and the weighted FTS spectrum (by the SRF of the radiometer) can help for instability correction, but what means (line 28) 'radiometric corrected FTS spectrum'? It should be better to present here the equation (that should be also a part of the main FTS measurement equation).

- Page 4, line 10: Maybe it should be indicated somewhere that the retrieval of I0 is 'model dependent' (in comparison with space measurements), through the relationship between SZA and the individual air mass, plus the modelization of the ozone layer.

- Page 4, line 12: the range of AMF was fixed to 1-3.5, instead of 2-6 or 2-8 in general. For AMF > 3.5 it is due to SNR limitation, we assumed, but what about the atmospheric instabilities for AMF below 2 (noon time at IZO)?

- Page 4, equation (3): so, on the left, there is all parameters/variables measured or estimated by calculation and modelling. But (line 21-23), using the AERONOET data from IZO and calculation of m_aod, it is also possible estimate the product Tau_aod x m_aod and to put this contribution on the left before the Langley regression. Is it how you proceeded?

- Page 4, line 22: write 'to the total optical depth' instead of 'to the aerosol optical depth'?

- Page 4: could it be possible to have at least, one Langley plot (one of the 7 selected) to improve the traceability of your works?

- Page 5, line 9 to 14: so you performed simulated Langley plots? Is it correct? What was the reference spectrum I0 used for this study?

- Page 7, line 1: could it be possible to add the reference for the kittPeak high resolution spectrum in absolute level?

- Page 9: comparison with other spectra. Even if the SSI variability is not very high above 300 nm, it should be preferably mentioned in the discussion that the comparisons are performed for the different dates (please indicate the dates) and thus, different solar activity. This fact could affect or not (depending on the amplitude of expected SSI changes) the ratio of spectra. Could you estimate this contribution?

*Technical corrections*

- Page 2, line 28: 'W' instead of 'S' for the longitude of IZO.

---

## Author Comment (AC1) · 20 Jun 2017

We would like to thank the referee for his positive review and the useful comments which we have addressed in the revised manuscript.

- The full field of view of QASUME is 2.5° ($\pm$1.25°), while it is 7° ($\pm$3.5°) for the FTS.
- QASUME acquired one spectrum every 15 minutes, while the FTS obtained one spectrum every 50 seconds.
* * *

---

## Author Response (AR1)

Dear editor,
Please find our response to the reviewers below, as well as the changes as found in the pdf markup document.

**Referee 1:**

**p.3, ln. 3: it would be better to use instead of 'resolution' the more specific term 'full width at half maximum'.**
Done : p3, line 3

**p.3, ln.7 and ln. 20: the field of view should be stated both times either with '+-' or with the full opening angle.**
Done: p3, line 8

**p.3: for both instruments, what was the frequency of the measurements, how many data points are available for the fit on each half day?**
Response: QASUME acquired one spectrum every 15 minutes, while the FTS obtained one spectrum every 50 seconds.
Done: p3, line 14-15 and p3, line 25.

**p.7, Fig. 3: the scales of the irradiance should be given for the 3 panels.**
Done: p8, figure 4.
* * *
**Referee 2:**

**Page 3, line 8: description of the tube for direct solar measurements (FOV of 2.5°). Is there any estimation of the circumsolar contribution (2.5° minus the solar disk) that should be non-negligible in the UV during field measurements (in comparison to space measurements)? At the PTB, the straylight in this FOV should have been normally removed during the calibration. Maybe the stability and performances of the solar tracker allowed a possible reduction of the FOV to ~1°? It is an important topic for the extra-terrestrial SSI retrieval from Langley plots.**
Response: The referee raised an important point concerning the contribution of scattered radiation in the field of view (FOV) of the instrument. Since the measurements took place at high elevation with low aerosol optical depth, the fraction of diffuse radiation scattered into the FOV of the instrument is expected to be very small. We have now modelled this contribution for the FOV of QASUME with the radiative transfer model libRadtran using as basis for the radiative transfer calculations the atmospheric conditions found at Izana, namely atmospheric pressure of 772 mbar and aerosol optical depth of 0.02 with an assumed asymmetry factor of 0.76. The model calculations were carried out for several airmasses between 1 and 3.5 and for the wavelength range between 290 nm and 500 nm, in order to simulate the effect of the scattered radiation on the Langley-plot retrieval. The contribution of scattered radiation in the FOV of QASUME with respect to the direct irradiance is largest at short wavelengths and large airmasses: at an airmass of 3.5 the contribution is 0.55% and 0.14% at 300 nm and 310 nm respectively. At smaller airmasses and longer wavelengths, this contribution decreases to less than 0.1%. The significant dependence on

wavelength suggests that the dominant contribution comes from Rayleigh scattering, while the forward scattered radiation from the aerosols is negligible.

Finally, we have included the modelled scattered radiation in one of the Langley-plot retrievals of QASUME to determine its effect on the retrieved zero-airmass irradiance: The resulting zero-airmass extrapolation differs by less than 0.1% from the original retrieval, demonstrating that scattered radiation into the FOV of QASUME for the condition encountered at Izana during the measurements has a negligible effect on the retrieved extraterrestrial solar spectrum.

While this effect is expected to be approximately a factor of 3 larger for the FTS due to its larger FOV with respect to QASUME, we have not taken it explicitly into account since any resulting bias in the retrieved solar spectrum of the FTS will be slowly varying in wavelength and will be implicitly taken into account when the QASUME and FTS spectra are combined, as explained in section 2.4.

Changes in manuscript:P8, lines 12-22 and P9, lines 1-8.

***Page 3, line 14: only one lamp was used to monitor a possible change of absolute responsivity due to transportation and aging effects. Why not more than one lamp? A triplet of lamps would have improved the uncertainty budget.***

Response: The calibration of QASUME during the field measurements was based on a set of three 250 W tungsten halogen lamps as described in Gröbner et al.,2005, which can be mounted in the portable field calibrator.

Changes in manuscript:P3, line 15-16.

***Page 3, line 20: same remark for the FOV and circumsolar UV contribution, fully negligible?***

Response: Please see answer above.

***Page 3, lines 25-29: it could be better for the reader to present the equation of this instability correction, instead of sentences. We understand that a comparison between the filter radiometer and the weighted FTS spectrum (by the SRF of the radiometer) can help for instability correction, but what means (line 28) 'radiometric corrected FTS spectrum'? It should be better to present here the equation (that should be also a part of the main FTS measurement equation).***

Response: This was a comment from the initial review and therefore already included in the initially revised manuscript (see equation 1, p 4).

***Page 4, line 10: Maybe it should be indicated somewhere that the retrieval of I0 is 'model dependent' (in comparison with space measurements), through the relationship between SZA and the individual air mass, plus the modelization of the ozone layer.***

Response: This is described in section 2.3 and section 2.4, and the uncertainties of the retrieval are described in detail. We believe that this is sufficient to describe the procedure applied to retrieve the extra-terrestrial spectrum.

*Page 4, line 12: the range of AMF was fixed to 1-3.5, instead of 2-6 or 2-8 in general. For AMF > 3.5 it is due to SNR limitation, we assumed, but what about the atmospheric instabilities for AMF below 2 (noon time at IZO)?*

Response: The airmass range used at Izana for the Langley-plot was in fact between 1.1 and 3.5 (lowest SZA was 27°). We have not encountered any atmospheric instabilities at noon during our measurements, maybe due to the fact that in September the atmospheric conditions were more stable than in the summer months when the sun reaches lower solar zenith angles.

Changes in manuscript: page 4, line 22

*Page 4, equation (3): so, on the left, there is all parameters/variables measured or estimated by calculation and modelling. But (line 21-23), using the AERONOET data from IZO and calculation of m_aod, it is also possible estimate the product Tau_aod x m_aod and to put this contribution on the left before the Langley regression. Is it how you proceeded?*

Response: As described in the text, we have only taken into account the Rayleigh extinction component in equation 4 and have refrained from removing the ozone and aerosol component. The reason is that the atmospheric pressure was known very well and did not vary significantly during each half-day. In contrast, the observed variability of the total column ozone and the aerosol optical depth measurements from co-located instruments resulted partly from instrumental noise (for example ±1 DU in individual total column ozone measurements from the Brewer spectrophotometer) so that we did not want to introduce artificial noise from other instruments in the Langley-plot retrievals of QASUME.

*Page 4, line 22: write 'to the total optical depth' instead of 'to the aerosol optical depth'?*

Response: This was also a comment from the initial review, see page 5, line 7.

*Page 4: could it be possible to have at least, one Langley plot (one of the 7 selected) to improve the traceability of your works?*

Done: page 5, figure 1, and new text at lines 1-5.

*Page 5, line 9 to 14: so you performed simulated Langley plots? Is it correct? What was the reference spectrum I0 used for this study?*

Response: The Langley-plot simulations to determine the contribution from the finite resolution of QASUME were performed with a high resolution spectrum based on KittPeak combined with the solar spectrum of Thuillier for the absolute irradiance level.

Changes in document: page 7, line 2-3

*Page 7, line 1: could it be possible to add the reference for the kittPeak high resolution spectrum in absolute level?*

Done.

Changes in document : page 7, line 18.

*Page 9: comparison with other spectra. Even if the SSI variability is not very high above 300 nm, it should be preferably mentioned in the discussion that the comparisons are performed for the different dates (please indicate the dates) and thus, different solar activity. This fact could affect or*

*not (depending on the amplitude of expected SSI changes) the ratio of spectra. Could you estimate this contribution?*

Response: The last point raised by the referee concerning the dates when the different SSI were measured is important and can be found in the respective references for each solar spectrum. The SORCE spectrum used here was measured on 27 November 2004 and the Sciamachy spectrum on 27 February 2003. SOLSPEC-A represents the average over the period 6 June 2008 to 29 April 2009 while SOLSPEC-B represents the average for the period April to June 2008. Finally the ATLAS-3 Spectrum (Thuillier) was measured on several days in November 1994. Thus all solar spectra shown here , including QASUME were either measured during solar minimum or in the declining phase of the respective solar cycle.

Current studies provide only an upper limit of the SSI variability throughout a solar cycle of the order of 1% to 2% due to the low solar variability in this wavelength region. Based on the results from SORCE SIM discussed in Harder et al., 2009, the variability of SSI between 300 nm to 500 nm is at most 0.5%. In that respect, the conclusions drawn from the comparisons shown in Figure 5 of the solar spectra from SORCE SIM, Sciamachy, ATLAS-3 from Thuillier, SOLSPEC and QASUME should not be significantly affected from being measured on different dates.
Changes in manuscript: page 10, lines 22-30.

We also performed some minor technical corrections to the manuscript such as correct subscript syntax.

[revised manuscript text omitted]